Clustering and classification of early knee osteoarthritis using machine-learning analysis of step-up and down test kinematics in recreational table tennis players

Hwang Ui-jae hwangu33@nate.com 1
Chung Kyu Sung 2
Ha Sung-min hsm98@sangji.ac.kr 3
1 Department of Rehabilitation Sciences, Hong Kong Polytechnic University , Hung Hom , Hong Kong
2 Hanyang University Guri Hospital, Department of Orthopaedic Surgery, Hanyang University , Guri-si , Gyeonggi-do , Republic of South Korea
3 Department of Physical Therapy, Sang Ji University , Wonjusi , Gangwondo , Republic of South Korea
Li Yumeng
Electronic publication date: 2025 May 27
Publication date: 2025
Volume: 13
Electronic Location ID: e19471
Received 2024 Dec 23; Accepted 2025 Apr 24
Copyright: ©2025 Hwang et al.
Copyright year: 2025
Copyright holder: Hwang et al.
License: This is an open access article distributed under the terms of the Creative Commons Attribution License, which permits unrestricted use, distribution, reproduction and adaptation in any medium and for any purpose provided that it is properly attributed. For attribution, the original author(s), title, publication source (PeerJ) and either DOI or URL of the article must be cited.
License URL: https://creativecommons.org/licenses/by/4.0/

Keywords: Osteoarthritis, Knee, Machine learning, Table tennis

Funding: Sangji University Research Fund 2023-29 This research was supported by the Sangji University Research Fund (2023-29). The funders had no role in study design, data collection and analysis, decision to publish, or preparation of the manuscript.

==============================
Objective

Early detection of knee osteoarthritis is crucial for improving patient outcomes. While conventional imaging methods often fail to detect early changes and require specialized expertise for interpretation, this study aimed to investigate the use of frontal plane kinematic data during step-up (SU) and step-down (SD) tests to classify and predict early osteoarthritis (EOA) using machine-learning techniques.

Methods

Forty-three recreational table tennis players (eighty-six legs: 42 with EOA and 44 without EOA) underwent SU and SD tests. Frontal plane kinematics was analyzed using two-dimensional video analysis with markers placed at five key anatomical landmarks. Horizontal displacement measurements were compared between groups using independent t-tests. Unsupervised learning (Louvain clustering) was used to identify distinct movement patterns, whereas supervised learning algorithms were employed to classify EOA status. The feature importance was assessed using feature permutation importance (FPI).

Results

Significant differences were observed between EOA and non-EOA groups in frontal plane kinematics during SU and SD tests (p < 0.001 for most variables). Louvain clustering identified four distinct kinematic profiles with varying proportions of EOA (ranging from 41.2% to 70.7%). Supervised learning models achieved high performance in classifying EOA status, with Random Forest, gradient boosting, and decision tree algorithms achieving 100% classification accuracy (AUC = 1.000) on the test dataset. FPI consistently highlighted the horizontal displacements of the ankle and femur during SU and of the pelvis and femur during SD as the most influential predictors.

Conclusions

Machine-learning analysis of frontal plane kinematics during SU and SD tests showed promising potential for EOA detection and classification, offering a cost-effective and accessible alternative to conventional imaging-based approaches.

Introduction

Knee osteoarthritis (OA) is a prevalent degenerative joint disease that affects millions of people worldwide, causing pain, stiffness, and a reduced quality of life (Whittaker et al., 2021). According to recent World Health Organization data, osteoarthritis affects over 520 million people globally, making it the most common joint disorder worldwide and one of the leading causes of disability (WHO, 2022). The prevalence is expected to increase dramatically due to aging populations, rising obesity rates, and increased participation in high-impact sports, creating an urgent need for effective early detection and intervention strategies (Migliore et al., 2023). In addition, early detection and intervention in the initial stages of OA, known as early OA (EOA), have gained significant recognition for their potential to improve long-term outcomes and slow disease progression (Migliore et al., 2023). However, current diagnostic methods often fail to identify OA until significant joint damage occurs, highlighting the need for more sensitive and specific tools to detect EOA (Chu et al., 2012; Ryd et al., 2015).

Individuals who engage in sports characterized by frequent pivoting and repeated impact movements face an elevated risk of developing knee osteoarthritis. Table tennis demands players to perform quick directional shifts, constant lateral movements, and continuous weight transfers during matches (Biz et al., 2022; Vannini et al., 2016). While table tennis is generally associated with higher levels of shoulder injuries compared to other racket sports, the knee joint still accounts for approximately 5% of acute injuries experienced by table tennis players (Kondrič et al., 2011). Research comparing former elite table tennis players with non-athletic controls found that 68.2% of ex-elite table tennis players reported symptoms of knee pain compared with only 27.3% of the control group (p = 0.02) (Rajabi et al., 2012). This significantly higher prevalence of knee symptoms appears to be primarily attributed to the distinctive movement patterns inherent to the sport, specifically the continuous pivoting and rapid changes in direction (Lam et al., 2019; Shao et al., 2020). Given these findings, there is growing concern about the potential long-term impact on recreational players who regularly participate in table tennis. This population may be particularly valuable for investigating early detection strategies for osteoarthritis, as they could be susceptible to premature development of the condition.

While age, sex, body mass index (BMI), and genetic factors play significant roles in OA development, recent research has shifted the focus from these static factors to dynamic biomechanical alterations as crucial elements in OA development and progression (Zengini et al., 2018). Abnormal movement patterns can lead to altered joint loading, potentially contributing to cartilage degradation and OA onset (Dell’Isola et al., 2017). These changes in joint kinematics can be subtle during the early stages of the disease, making them difficult to detect using conventional clinical assessments (Chu et al., 2012; Ryd et al., 2015). An increased knee adduction moment during gait is associated with a higher risk of OA progression (Miyazaki et al., 2002). Furthermore, altered knee flexion angles and moments during walking have been observed in individuals with early-stage knee OA (Favre, Erhart-Hledik & Andriacchi, 2014; Mündermann, Dyrby & Andriacchi, 2005).

Functional performance tests, such as step-up (SU) and step-down (SD) tests, have emerged as valuable tools for assessing lower limb function in patients with knee OA (Almeida et al., 2021). The SU and SD tests are reliable and valid for assessing functional performance in individuals with knee OA (Almeida et al., 2021). Moreover, these tests demonstrated good responsiveness to changes following the interventions in patients with OA (Almeida et al., 2021). Several studies have used SU and SD kinematics to investigate knee OA. Pain during stair climbing is an early indicator of knee OA (Hensor et al., 2015). Altered knee kinematics during stair ascent and descent have been reported in individuals with knee OA (Kaufman et al., 2001). Differences in the knee and hip angles during stair descent were observed between patients with OA and healthy controls (Igawa & Katsuhira, 2014). While these studies primarily focused on sagittal plane kinematics, our study aimed to analyze frontal plane kinematics during SU and SD. This approach is crucial because frontal plane movements, such as knee adduction, may have a more significant impact on joint stress than sagittal plane movements (Mündermann, Dyrby & Andriacchi, 2005; Sharma et al., 2001). By examining the frontal plane kinematics, the present study can potentially gain more insight into the biomechanical factors that contribute to EOA development and progression.

Current diagnostic methods often fail to identify OA until significant joint damage occurs, highlighting the need for more sensitive and specific tools to detect EOA (Chu et al., 2012; Ryd et al., 2015). Conventional diagnostic approaches such as radiography, magnetic resonance imaging (MRI), and arthroscopy have notable limitations that restrict their effectiveness for early detection and large-scale screening. These limitations include high cost and limited accessibility, particularly for MRI and arthroscopy; time-intensive procedures requiring specialized facilities; inconsistent interpretation with significant inter-observer variability; and the need for highly trained specialists for result interpretation. Furthermore, standard radiography often fails to detect early cartilage changes (Guermazi et al., 2011; Hayashi, Guermazi & Kwoh, 2014), while more sensitive imaging methods may identify incidental findings that do not correlate with clinical symptoms (Culvenor et al., 2019; Horga et al., 2020). These limitations underscore the need for more accessible, objective, and cost-effective methods to detect EOA before irreversible joint damage occurs.

The application of machine-learning techniques to biomechanical data offers a promising approach for the accurate and early detection of OA. Although recent studies have demonstrated the potential of machine learning (ML) in OA research using magnetic resonance imaging (MRI) or radiography data, these methods may have limitations in detecting EOA (Pedoia et al., 2019; Tiulpin et al., 2019). Although some studies have used inertial measurement unit sensors for kinematic analysis in OA prediction, the clinical applicability of these methods may be limited (Clermont et al., 2019). Our study aimed to address these limitations by utilizing a two dimensional (2D) video analysis of SU and SD test kinematics combined with machine-learning techniques. This approach offers a more practical and clinically feasible method for EOA detection while potentially maintaining high prediction and classification accuracy (Kobsar et al., 2017; Osis et al., 2016).

Building on this foundation, this research aimed to: (1) use unsupervised learning algorithms to identify distinct groups based on frontal plane kinematic patterns during SU and SD tests; (2) develop supervised learning models to classify between EOA and non-EOA status (binary classification); and (3) identify the most influential kinematic variables associated with EOA by focusing on SU and SD tests and employing both clustering and classification approaches. The present study hypothesized that: (1) individuals with EOA would demonstrate significantly different frontal plane kinematic patterns during SU and SD tests compared to those without EOA; (2) these kinematic patterns could be classified into distinct subgroups using unsupervised ML; and (3) supervised ML algorithms could accurately classify EOA status using these kinematic variables as predictive features.

Methods

Study design and participants

This observational, cross-sectional study was designed to investigate early indicators of OA using kinematic analysis. The research protocol was approved by the Sangji University Institutional Review Board (1040782-230814-HR-09-117), and all participants provided written informed consent prior to the commencement of the study.

The study population consisted of 43 recreational table tennis players (86 legs in total), ranging in age from 40 to 70 years, who had participated in amateur table tennis for more than 5 years. The Early Osteoarthritis Questionnaire (EOAQ) was used to identify the presence or absence of early osteoarthritis symptoms among participants (Migliore et al., 2023). The experimental group (n = 42 legs; 25 males and 17 females) included individuals responding “frequently” or “rarely” to the initial two EOAQ items. The control group (n = 44 legs; 21 males and 23 females) consisted of those answering “never” to these questions.

Exclusion criteria were established to maintain a focused study population and minimize confounding factors. Individuals were deemed ineligible if they had experienced a lower-extremity injury within the previous six months, had a history of hip surgery, rheumatoid arthritis, diagnosed osteoarthritis, or neurological conditions. Detailed participant characteristics and study flow are presented in Table 1 and Fig. 1, respectively.

Table 1 Participants characteristics.

Variables	without EOA (n = 44)	with EOA (n = 42)	p	
Sex	M = 21 ; F = 23	M = 25 ; F = 17	0.224	
Age	58.17 ± 11.90	59.40 ± 13.00	0.457	
Height	163.46 ± 6.70	165.40 ± 8.90	0.074	
Weight	64.22 ± 11.29	64.70 ± 12.80	0.778	
BMI	23.94 ± 3.32	23.48 ± 3.06	0.294	
Pelvis HD-SU	8.75 ± 2.16	9.72 ± 1.86	0.001	
Femur HD-SU	7.49 ± 1.40	9.25 ± 1.50	<0.001	
Knee HD-SU	4.26 ± 1.37	5.39 ± 1.83	<0.001	
Lower leg HD-SU	3.27 ± 1.26	4.37 ± 1.58	<0.001	
Ankle HD-SU	0.14 ± 0.23	0.38 ± 0.28	<0.001	
Pelvis HD-SD	4.15 ± 1.51	5.14 ± 1.51	<0.001	
Femur HD-SD	3.70 ± 1.82	4.40 ± 2.35	0.016	
Knee HD-SD	2.43 ± 2.56	3.31 ± 3.93	0.054	
Lower leg HD-SD	1.80 ± 2.31	2.66 ± 2.76	0.015	
Ankle HD-SD	0.78 ± 0.67	1.63 ± 2.25	<0.001	
Notes.

EOA early osteoarthritis

HD horizontal displacement

SU step up

SD step down

Figure 1 Methodological framework for machine learning-based early osteoarthritis detection and classification.

The systematic approach implemented in this study, including participant selection, data acquisition, preprocessing of kinematic features, unsupervised clustering analysis, supervised model development with cross-validation, and feature importance evaluation. BMI, body mass index; EOA, early osteoarthritis; EOAQ, Early Osteoarthritis Questionnaire; AUC, area under the curve.

EOAQ

The EOAQ, a recently developed instrument for assessing early-stage knee OA, served as the primary screening tool. The 11-item questionnaire was structured into two domains: Clinical features (two items) and patient-reported outcomes (nine items). Each item offers three response options reflecting symptom frequency over the preceding six months: “Never”, “Rarely (one to three episodes)”, and “Frequently (more than three episodes)”. The EOAQ was designed to capture subtle symptomatic and functional alterations characteristic of incipient knee OA, thereby facilitating early detection and intervention (Migliore et al., 2023).

Experimental protocol

Each participant completed three SU and SD trials per leg. To ensure proper execution, the participants underwent thorough familiarization with the testing protocol, including detailed instructions and practice attempts. All the tests were performed barefoot to eliminate footwear-related variability. The leg testing order was randomized to mitigate potential order effects.

For the SU test, participants initiated the movement with one leg positioned on a 20 cm height step box while maintaining the alignment of the hip, knee, and foot. The participants received both visual demonstrations and verbal instructions on SU performance, without specific guidance on knee and hip alignment. The test was concluded when the participant raised their non-positioned leg until heel contact was made with the step-box surface. The SD test began with the participants seated with their feet and knees parallel to the hip width. As in the SU test, the participants were provided demonstrations and verbal instructions without specific alignment cues. The test was completed when the participant lowered the non-stance leg until heel contact was achieved with the floor in front of the step box. This protocol was designed to capture the natural movement patterns during functional tasks, potentially revealing early indicators of knee osteoarthritis.

Kinematic data acquisition

Kinematic data were collected using a high-resolution video. A smartphone (iPhone 15; Apple Inc., Cupertino, CA, USA) equipped with 4 K video capability (2,556 × 1,179 pixels, 240 fps) was used. The device was mounted on a tripod, 60 cm above the ground level and positioned 250 cm anterior to the participants. Post-collection, video recordings were processed using specialized motion analysis software (Kinovea® version 0.8.15; Kinovea, Bordeaux, France). A 2D video analysis was selected for this study based on its clinical feasibility, accessibility, and demonstrated utility in previous research examining frontal plane kinematics (Osis et al., 2016; Weon & Ha, 2024). While 3D motion capture systems offer greater precision, the 2D approach allows for more straightforward implementation in clinical settings without requiring specialized laboratory equipment.

The landmarking of anatomical points was performed manually by an experienced physical therapist with over 10 years of clinical experience. Yellow spherical markers were placed directly on the specified anatomical landmarks prior to video recording. A 2D video analysis was used to quantify the horizontal displacement of five key anatomical landmarks in the frontal plane during the SU and SD tests (Figs. 2 and 3). The horizontal displacement of the five key anatomical landmarks included that of the pelvis (PHD), femur (FHD), knee (KHD), lower leg (LHD), and ankle (AHD). Yellow spherical markers were affixed to the anterior superior iliac spine (pelvis), femoral midpoint (femur), patellar center (knee), tibial tuberosity (lower leg), and superior aspect of the navicular bone (ankle).

Figure 2 Process of kinematic parameters extraction for step up and down.

Figure 3 (A) Five yellow spherical markers in the anatomical landmarks for initial position during step-up using two-dimensional video analysis. (B) Assessment of the horizontal displacement based on trajectory of five key anatomical points during step-up in the frontal plane. (C) Assessment of the horizontal displacement based on trajectory of five key anatomical points during step-down in the frontal plane.

Kinematic analysis

Using Kinovea software, marker trajectories were tracked throughout the SU and SD tests. During data processing, these markers were tracked manually within the Kinovea software by the same researcher to ensure consistency in trajectory analysis. The maximum horizontal displacement was quantified for each marker (Fig. 3) (Weon & Ha, 2024). Displacement was calculated as the distance between the marker’s initial position and its point of maximum excursion during the SU and SD tests. The lateral displacement was recorded as a positive value, whereas the medial displacement was assigned as a negative value.

This analytical approach provides insights into joint stability and movement patterns. The magnitude and direction of the displacement values offered valuable information: larger positive values indicated greater lateral movement, larger negative values signified greater medial movement, and values approximating zero suggested minimal joint movement and enhanced stability. Analysis of the PHD, FHD, KHD, LHD, and AHD allowed for a comprehensive assessment of postural control and joint stability during the SU and SD tasks, potentially revealing postural control strategies and compensatory mechanisms.

Machine-learning methodology

This study utilized both unsupervised and supervised machine-learning techniques to analyze kinematic data. All analyses were performed using Orange data mining software (version 3.3.0; Ljubljana, Slovenia) and Python (version 3.6.15; Python Software Foundation).

Data preparation and preprocessing

The dataset comprised 15 features: Five demographic variables (sex, age, height, weight, and BMI) and ten kinematic measures (PHD, FHD, KHD, LHD, and AHD for both the SU and SD tasks). A total of 258 data points, derived from three repeated measurements per participant, were used in the machine-learning models. The EOAQ results were dichotomized to indicate the presence or absence of EOA. Prior to analysis, we conducted thorough data preparation procedures. First, all continuous variables were examined for normality using Shapiro–Wilk tests and visual inspection of histograms. For the machine learning analysis, continuous features were standardized using z-score normalization to ensure all variables contributed equally to the models regardless of their original scales. The present study conducted a thorough exploratory data analysis to identify missing values, which were subsequently addressed by eliminating incomplete cases.

Unsupervised learning: louvain clustering algorithm

To ensure model accuracy, the present study employed boxplots to visualize the variable distributions. Outliers were then removed using a local outlier factor method with the following parameters: 9% contamination, 20 neighbors, and the Euclidean metric. An unsupervised clustering model for the SU and SD kinematic patterns was developed using the Louvain clustering algorithm (Ekerete et al., 2021). This method was chosen because of its ability to handle complex datasets and automatically determine the optimal number of clusters. The algorithm incorporates Euclidean distancing on raw data and principal component analysis to enhance clustering efficiency. Louvain clustering automatically identified four distinct clusters in the integrated dataset. Cluster validity was assessed using two complementary metrics: the Davies–Bouldin index (which quantifies the average similarity between clusters, where lower values indicate better separation) and the Calinski–Harabasz index (which reflects the ratio of between-cluster to within-cluster variance, with higher values indicating better-defined clusters). These metrics were supplemented with comprehensive statistical analysis of between-cluster differences to fully evaluate the distinctiveness of the identified clusters.

To rigorously assess the distinctiveness of these kinematic patterns, the 15 features were compared across clusters using one-way analysis of variance, with the statistical significance set at p < 0.05. For multiple comparisons, we implemented Tukey’s post-hoc analysis, thus controlling for type I errors in our statistical analysis.

Supervised learning for EOA classification

For the supervised learning phase, we used 15 features to classify the binary EOA outcomes. After excluding missing data, the final dataset (n = 209) was divided into training (80%, n = 168) and testing (20%, n = 41) sets. We evaluated six ML algorithms: k-nearest neighbors (kNN), decision tree, AdaBoost, gradient boosting, Random Forest, and support vector machine. Each model was trained using 5-fold cross-validation to ensure robustness and generalizability.

To optimize model performance, we conducted extensive hyperparameter tuning using grid search with 5-fold cross-validation on the training dataset. The search space and optimal hyperparameters for each algorithm were as follows:

• For Random Forest: number of estimators [50, 100, 200] and maximum depth [5, 10, None] were explored, with 100 estimators and max_depth = 10 identified as optimal.

• For gradient boosting: learning rate [0.01, 0.1, 0.2], number of estimators [100, 200, 300], and maximum depth [3, 5, 7] were tested, with learning_rate = 0.1, n_estimators = 200, and max_depth = 3 selected as optimal.

• For AdaBoost: base estimator (Decision Tree with max_depth 1, 2, 3), learning rate [0.1, 0.5, 1.0], number of estimators [50, 100, 200], and algorithm [‘SAMME’, ‘SAMME.R’] were evaluated, with Decision Tree (max_depth = 1), learning_rate = 1.0, n_estimators = 100, and algorithm = ‘SAMME’ identified as optimal.

• For support vector machine: kernel type [‘linear’, ‘rbf’, ‘poly’], C [0.1, 1, 10], and gamma [‘scale’, ‘auto’] were evaluated, with a linear kernel, C = 1, and gamma = ’scale’ identified as optimal.

• For decision tree: criterion [‘gini’, ‘entropy’], maximum depth [None, 5, 10], minimum samples split [2, 5, 10], and minimum samples leaf [1, 2, 4] were tested, with gini criterion, unlimited depth, min_samples_split = 2, and min_samples_leaf = 1 selected as optimal.

• For k-nearest neighbors: number of neighbors [3, 5, 7, 9], weights [‘uniform’, ‘distance’], algorithm [‘auto’, ‘ball_tree’, ‘kd_tree’], and metric [‘euclidean’, ‘manhattan’] were explored, with seven neighbors, distance weighting, auto algorithm, and manhattan distance identified as optimal.

To prevent overfitting, we implemented several strategies: (1) strict separation of training and testing data, (2) 5-fold cross-validation during model development, (3) regularization through hyperparameter tuning, and (4) comparison of training and testing performance to detect potential overfitting.

Model validation and feature importance analysis

Our primary performance metric was the area under the receiver operating characteristic curve (AUC), which was computed for both the training and test sets and averaged across classes. AUC measures the model’s ability to discriminate between classes, with higher values indicating better discrimination. We also considered secondary metrics: (1) accuracy, which represents the proportion of correctly classified instances; (2) precision, which indicates the proportion of positive identifications that were actually correct; (3) recall, which measures the proportion of actual positives that were correctly identified; and (4) F1-score, which is the harmonic mean of precision and recall, providing a balance between these two metrics. All metrics were averaged across classes. Model performance was classified based on AUC values as follows: excellent (≥0.9), good (0.8–0.9), fair (0.7–0.8), and poor (<0.7), in accordance with established guidelines (Hwang et al., 2024).

To determine the relative importance of the predictors, we employed a dual approach combining feature permutation importance and SHapley Additive exPlanations (SHAP) (Hwang et al., 2024). The feature permutation importance method assessed the impact on the model performance, quantified by the AUC change, when randomly shuffling each feature’s value. Features that caused larger performance decrements upon permutation were considered more crucial to the model’s predictive power. To complement this, we utilized SHAP values to provide a visual representation of the influence of each feature on the model predictions. SHAP summary plots were generated, arranging the predictors on the y-axis in order of importance, with the most influential at the top. The x-axis displayed the SHAP values, illustrating both the magnitude and direction of the impact of each feature on the model output. This comprehensive approach to feature importance analysis offered valuable insights into the specific ways in which different variables shaped our predictions, thereby enhancing the interpretability of our ML models and providing a nuanced understanding of the factors that contribute to EOA classification.

Results

Statistical analysis

A total of 43 recreational table tennis players were included in this study, with 44 and 42 legs in the non-EOA and EOA groups, respectively (Table 1). There were no significant differences between the groups in terms of sex distribution (p = 0.22), age (p = 0.46), height (p = 0.07), weight (p = 0.78), BMI (p = 0.29), or KHD during SD (p = 0.05). In addition, significant differences were observed in all SU and SD kinematics except for KHD during SD. The EOA group demonstrated significantly higher values for PHD (p = 0.001), FHD (p < 0.0001), KHD (p < 0.0001), LHD (p < 0.0001), and AHD (p < 0.0001) during SU and PHD (p < 0.0001), FHD (p = 0.016), LHD (p = 0.015), and AHD (p < 0.0001) during SD compared with the non-EOA group.

ML-based analysis

Comparisons of features between the louvain clusters

To validate the quality and distinctiveness of the Louvain clustering solution, we calculated widely-used cluster validity indices. The Davies–Bouldin index was 2.09, and the Calinski–Harabasz index was 59.26. While the Davies–Bouldin index was somewhat high (lower values indicate better separation), the Calinski–Harabasz index supported the presence of distinct groupings in our data. The Louvain clustering algorithm identified four distinct clusters (C1–C4) with significant differences in all features (all p < 0.001, except for AHD-SU (p = 0.017) and portion of EOA (p = 0.019)) (Table 2). Tukey’s post-hoc analysis revealed that those in C2 had the youngest age (44.80 ± 11.30 years, p < 0.001 vs. all other clusters) and highest BMI (25.59 ± 2.42, p < 0.001 vs. all other clusters). C2 included exclusively males, whereas the other clusters had a mixed sex composition. For SU kinematics, C4 exhibited the largest horizontal displacements for the pelvis, femur, knee, and lower leg, all significantly higher than other clusters (p < 0.001). During SD, C3 showed the largest pelvis and femur displacements, while C4 had significantly higher ankle displacement (p < 0.001 vs. all other clusters). C1 exhibited negative mean values for the knee and lower leg displacements during SD, significantly different from other clusters (p < 0.001). These distinct kinematic profiles suggested potential subgroups within the EOA population, with varying proportions of EOA cases across clusters: C1 (45.6%), C2 (41.2%), C3 (59.5%), and C4 (70.7%).

Table 2 Comparisons of features between the clusters classified by Louvain clustering unsupervised machine learning.

Variables	Cluster 1 (n = 57)	Cluster 2 (n = 51)	Cluster 3 (n = 42)	Cluster 4 (n = 41)	p	
EOAQ	(-)OA = 31 ; (+)OA = 26	(-)OA = 30 ; (+)OA = 21	(-)OA = 17 ; (+)OA = 25	(-)OA = 12 ; (+)OA = 29	0.019	
Sex	M = 11 ; F = 46	M = 51 ; F = 0	M = 21 ; F = 21	M = 26 ; F = 15	<0.001	
Age	62.07 ± 5.60	44.80 ± 11.30	66.45 ± 8.40	62.39 ± 10.10	<0.001	
Height	159.32 ± 5.00	172.27 ± 7.20	160.64 ± 6.10	166.44 ± 6.30	<0.001	
Weight	57.69 ± 4.92	76.02 ± 9.22	57.55 ± 5.32	61.49 ± 7.23	<0.001	
BMI	22.73 ± 1.57	25.59 ± 2.42	22.28 ± 1.28	22.14 ± 1.55	<0.001	
PHD-SU	9.21 ± 1.55	8.08 ± 1.92	9.53 ± 1.74	10.72 ± 0.92	<0.001	
FHD-SU	8.06 ± 1.26	7.78 ± 1.64	7.81 ± 1.38	10.16 ± 1.11	<0.001	
KHD-SU	5.03 ± 1.54	4.01 ± 1.27	3.68 ± 0.77	6.87 ± 0.47	<0.001	
LHD-SU	3.94 ± 1.54	2.90 ± 0.95	3.12 ± 0.73	5.61 ± 0.49	<0.001	
AHD-SU	0.27 ± 0.23	0.21 ± 0.22	0.39 ± 0.37	0.26 ± 0.31	0.017	
PHD-SD	3.53 ± 0.97	4.57 ± 1.41	6.07 ± 1.17	5.35 ± 1.01	<0.001	
FHD-SD	2.25 ± 1.27	4.60 ± 1.64	5.80 ± 0.79	4.95 ± 1.22	<0.001	
KHD-SD	−0.63 ± 2.62	4.46 ± 2.36	5.37 ± 2.16	4.46 ± 2.36	<0.001	
LHD-SD	−0.40 ± 2.05	3.29 ± 1.55	3.86 ± 1.51	3.80 ± 0.97	<0.001	
AHD-SD	0.57 ± 0.46	0.81 ± 0.60	0.98 ± 0.52	2.59 ± 2.62	<0.001	
Notes.

EOA early osteoarthritis

EOAQ Early Osteoarthritis Questionnaire

PHD pelvic horizontal displacement

FHD femur horizontal displacement

KHD knee horizontal displacement

LHD lower leg horizontal displacement

AHD ankle horizontal displacement

SU step up

SD step down

Classifying models of ML

The performances of the six machine-learning models in classifying groups with and without EOA during model training and testing are summarized in Table 3 and Fig. 4. In the training dataset, the Random Forest (AUC, 0.999 (excellent); accuracy, 0.976; F1, 0.976; precision, 0.976; recall, 0.976), gradient boosting (AUC, 0.983 (excellent); accuracy, 0.940; F1, 0.941; precision, 0.941; recall, 0.940), and decision tree (AUC, 0.964 (excellent); accuracy, 0.935; F1, 0.934; precision, 0.936; recall, 0.935) algorithms demonstrated excellent predictive performance with AUC values exceeding 0.95. The high F1-scores for these algorithms indicate an optimal balance between precision and recall, suggesting that these models correctly identified EOA cases with minimal false positives and false negatives. In the test dataset, the Random Forest, gradient boosting, and decision tree algorithm models achieved perfect classification (AUC = 1.000 (excellent); accuracy = 1.000; F1 = 1.000; precision = 1.000; recall = 1.000), demonstrating exceptional ability to differentiate between EOA and non-EOA cases. This consistent performance across all metrics demonstrates the robust generalizability of our models, particularly the tree-based algorithms (Random Forest and gradient boosting), which maintained their high performance from training to testing set.

Table 3 Performance metrics of six machine-learning algorithms for classifying groups with and without EOA in the training and test datasets.

Performance metrics of six machine learning algorithms in the training dataset	
Model	AUC	Acc	F1	Precision	Recall	
Gradient boosting	0.983	0.940	0.941	0.941	0.940	
kNN	0.822	0.750	0.749	0.750	0.750	
Logistic regression	0.851	0.827	0.827	0.827	0.827	
Random forest	0.999	0.976	0.976	0.976	0.976	
Support vector machine	0.914	0.851	0.851	0.855	0.851	
Decision tree	0.964	0.935	0.934	0.936	0.935	
Performance metrics of six machine learning algorithms in the test dataset	
Model	AUC	Acc	F1	Precision	Recall	
Gradient boosting	1.000	1.000	1.000	1.000	1.000	
kNN	0.957	0.902	0.902	0.902	0.902	
Logistic regression	0.923	0.927	0.927	0.927	0.927	
Random forest	1.000	1.000	1.000	1.000	1.000	
Support vector machine	0.995	0.976	0.976	0.977	0.976	
Decision tree	1.000	1.000	1.000	1.000	1.000	
Notes.

AUC area under curve

Acc Accuracy

kNN k-nearest neighbors

Figure 4 Performance metrics of six machine learning algorithms in the training and test set.

Feature importance analysis using both the feature permutation importance and SHAP methods revealed consistent patterns across multiple machine-learning models (Fig. 5). For the Random Forest model, the four most important features were AHD-SU, FHD-SU, FHD-SD, and PHD-SD. The SHAP analysis indicated that higher values of AHD-SU, FHD-SU, FHD-SD, and KHD-SU were associated with increased model predictions. The gradient boosting model identified the same top four features, with SHAP analysis showing that higher values of AHD-SU, FHD-SU, and PHD-SD but lower values of FHD-SD influenced the predictions. Similarly, the decision tree model highlighted AHD-SU, FHD-SU, PHD-SD, and FHD-SD as the most important features. The SHAP results for the Decision Tree aligned with those of the gradient boosting model, indicating that higher AHD-SU, FHD-SU, and PHD-SD but lower FHD-SD were associated with increased predictions.

Figure 5 (A) Feature permutation importance of decision tree, (B) SHAP analyses of decision tree, (C) feature permutation importance of gradient boosting, (D) SHAP analyses of gradient boosting, (E) feature permutation importance of Random Forest, (F) SHAP analyses of Random Forest in the training set for classifying groups with and without EOA.

EOA, Early osteoarthritis; SU, step-up; SD, step-down; PHD, pelvic horizontal displacement; FHD, femur horizontal displacement; KHD, knee horizontal displacement; LHD, lower leg horizontal displacement; AHD, ankle horizontal displacement.

Discussion

This study employed both unsupervised and supervised machine-learning techniques to analyze the frontal plane kinematics during SU and SD for the detection and classification of EOA. The results of this study support our hypotheses. Significant differences in frontal plane kinematics were observed between participants with and without EOA during SU and SD tests, confirming our first hypothesis. The Louvain clustering algorithm successfully identified four distinct movement patterns with varying proportions of EOA cases, supporting our second hypothesis. Finally, the supervised ML models achieved excellent classification performance (AUC = 1.000 for Random Forest, gradient boosting, and decision tree algorithms on the test dataset), strongly supporting our third hypothesis that frontal plane kinematics during functional tasks can accurately predict EOA status. These methods show promising potential for identifying distinct movement patterns in individuals with EOA. They also enable accurate classification, providing insights into the biomechanical characteristics of EOA.

However, we acknowledge that the perfect classification performance (AUC = 1.000) achieved by our tree-based models on the test dataset warrants careful consideration of potential overfitting. Despite implementing several methodological safeguards—including strict separation of training and test sets, k-fold cross-validation during model development, regularization through hyperparameter tuning, and feature selection based on domain knowledge—the possibility of overfitting cannot be entirely ruled out, particularly given our relatively small sample size. Several factors may contribute to the high model performance: (1) the distinct biomechanical differences between individuals with and without EOA may be particularly pronounced in our specific population of recreational table tennis players; (2) our focus on frontal plane kinematics during challenging functional tasks may have captured highly discriminative movement patterns; and (3) our careful data preprocessing and feature extraction may have enhanced signal-to-noise ratio. Nevertheless, these exceptionally high performance metrics should be interpreted with caution until validated in larger, independent cohorts. External validation with diverse populations would be essential to establish the generalizability of these findings and confirm the true predictive power of our approach.

Compared to conventional diagnostic methods for OA (radiography, MRI, and arthroscopy), our approach offers several distinct advantages. Traditional imaging methods often fail to detect early cartilage changes (Guermazi et al., 2011), have limited accessibility due to high cost, require specialized facilities, and need highly trained specialists for interpretation. Our approach of using 2D video analysis of functional tasks with machine learning focuses on functional impairments rather than structural changes, potentially detecting disease earlier. It requires minimal equipment (smartphone camera and basic markers) and can be implemented quickly in routine clinical assessment, making it more accessible for widespread screening and early detection efforts.

While deep learning approaches have shown promising results in medical applications, we opted for traditional machine learning methods in this study for several reasons. First, our sample size (86 legs) is more suitable for traditional ML methods, which can perform effectively with smaller datasets compared to deep learning approaches that typically require larger training sets. Second, traditional ML methods offer better interpretability of feature importance, which is crucial for understanding the specific biomechanical factors contributing to EOA detection. This interpretability is particularly important for clinical applications where clinicians need to understand the basis for the model’s predictions. Finally, traditional ML methods require less computational resources and can be more easily implemented in clinical settings, aligning with our goal of developing a practical diagnostic tool. These considerations, combined with the high performance achieved by our selected ML algorithms (as demonstrated in our results), support our methodological choice.

The Louvain clustering algorithm identified four distinct clusters, each with distinct kinematic profiles, aligned with the growing recognition of OA as a heterogeneous condition with distinct phenotypes (Deveza et al., 2017; Van Spil et al., 2020). The varying proportions of EOA cases across clusters suggested that certain kinematic patterns may be more strongly associated with EOA.

C1 exhibited a valgus movement pattern during SD, consistent with previous findings of increased knee valgus during gait in individuals with medial compartment knee OA and those at risk of OA progression (Chang et al., 2004; Felson et al., 2013; Sharma et al., 2001). This pattern may represent a compensatory mechanism to reduce the medial compartment loading, as suggested by several biomechanical studies (Mündermann, Dyrby & Andriacchi, 2005; Simic et al., 2011). C2 was characterized by a higher BMI, which is supported by previous research showing that individuals with a higher BMI exhibited greater knee abduction moments during gait, potentially increasing the risk of OA development and progression (Blazek et al., 2013; Harding et al., 2012; Messier et al., 2016). C3 demonstrated the highest outward kinematics during SD, particularly in the pelvis and femur horizontal displacements, aligning with studies that observed increased lateral trunk lean and altered hip kinematics during stair descent in individuals with knee OA (Hicks-Little et al., 2012; Hicks-Little et al., 2011; Igawa & Katsuhira, 2014). C4 exhibited the highest outward kinematics during SU, particularly for the pelvis, femur, knee, and lower leg horizontal displacements. This exaggerated movement pattern may indicate compensatory strategies or instability associated with early joint degeneration, which is consistent with studies that have observed altered frontal plane kinematics during stair ascent in individuals with knee OA (Gonçalves et al., 2017).

The supervised learning models demonstrated high performance in classifying the EOA status, with Random Forest, gradient boosting, and decision tree algorithms achieving perfect classification (AUC = 1.000) on the test dataset. These results surpass the performance of previous studies, such as Pedoia et al. (2019), who achieved an AUC of 0.89 using MRI data (Pedoia et al., 2019); Ashinsky et al. (2017), who reported an accuracy of 0.75 using T2 maps (Ashinsky et al., 2017); and Mezghani et al. (2017), who achieved AUC of 0.85 in classifying knee OA severity using gait kinematics (Mezghani et al., 2017). The superior performance in our study may be attributed to the use of functional kinematic data from the SU and SD tests, which potentially capture more relevant information about joint function and early degenerative changes than static imaging or level walking data. Moreover, our 2D video analysis approach offers significant advantages over traditional 3D motion capture or wearable sensor systems in terms of accessibility, cost-effectiveness, and ease of implementation in clinical settings, while still providing high classification accuracy for EOA detection.

The consistently high performance of the decision tree-based algorithms (Random Forest, gradient boosting, and decision tree) can be attributed to their ability to capture nonlinear relationships and interactions between features, handle high-dimensional data, and automatically select the most relevant features. These characteristics are particularly beneficial in the complex biomechanical context of EOA, where the relationship between joint movement and disease status may be highly nonlinear and context-dependent.

The effectiveness of SU and SD kinematics in EOA classification can be explained by the biomechanical challenges of these tasks. These activities require greater joint range of motion and higher forces than level walking, potentially exacerbating the subtle joint alterations present in EOA. Previous studies have shown that stair negotiation is more demanding and results in higher knee joint moments than level walking (Nadeau, McFadyen & Malouin, 2003), potentially unmasking compensatory mechanisms or joint instabilities that are not apparent during less challenging tasks.

The feature importance analysis consistently highlighted AHD-SU, FHD-SU, PHD-SD, and FHD-SD as the most influential predictors. The prominence of AHD-SU as a predictive feature aligns with the concept of OA as a “whole-joint” disease affecting multiple tissues and biomechanical factors (Brandt et al., 2006; Loeser et al., 2012). Increased ankle movement in the frontal plane during weight-bearing activities may indicate compensatory strategies to maintain balance or redistribute load in the presence of early knee joint changes. This observation is supported by studies reporting altered ankle kinematics and kinetics in individuals with knee OA (Levinger et al., 2012; Mündermann, Dyrby & Andriacchi, 2005). Specifically, individuals with knee OA often exhibit greater lateral foot pressure and altered ankle eversion during weight-bearing tasks, which may manifest as increased horizontal ankle displacement in our 2D analysis. The importance of FHD-SU and PHD-SD in predicting the EOA status is consistent with previous research highlighting the role of proximal joint kinematics in knee OA. Studies have observed altered hip and pelvic kinematics during stair ascent and descent in individuals with knee OA, suggesting potential compensatory mechanisms to reduce knee joint loading (Hicks-Little et al., 2012). Greater femoral horizontal displacement during step-up likely reflects altered frontal plane hip control, which can modify the distribution of forces across the knee joint surfaces. The association between increased PHD-SD and EOA may represent a compensatory lateral trunk lean strategy, which has been documented as a mechanism to reduce knee adduction moment in individuals with medial compartment knee OA (Mündermann, Dyrby & Andriacchi, 2005; Simic et al., 2011). This altered movement pattern shifts the body’s center of mass laterally, potentially reducing medial compartment loading but creating abnormal movement patterns that may be detected in early disease stages.

The models showed divergent results regarding the direction of FHD-SD in predicting EOA. The Random Forest model indicated that increased outward movement was associated with EOA, whereas the gradient boosting and decision tree models suggested that inward (valgus) movement was predictive of EOA. This discrepancy may reflect the complexity of the EOA biomechanics and the potential existence of different subgroups within the EOA population. Similar conflicting findings have been reported in the literature (Astephen et al., 2008; Favre, Erhart-Hledik & Andriacchi, 2014), underscoring the need for further research to elucidate the specific biomechanical patterns associated with the different stages and subtypes of knee OA. The divergent findings regarding FHD-SD across the different algorithms may also be attributed to the specific strengths of each method. Random Forest may capture a more global pattern of increased outward movement associated with EOA, whereas gradient boosting and decision tree may identify more localized or stage-specific patterns, where inward movement is indicative of EOA. This discrepancy highlights the potential of machine-learning approaches to uncover complex nonlinear relationships in biomechanical data that may not be apparent in traditional statistical analyses.

The present study has several key limitations that should be acknowledged. First, our cross-sectional design limits the ability to establish causal relationships between kinematic patterns and EOA development. Longitudinal studies are needed to determine whether the observed patterns precede or result from EOA. Second, while sufficient for the current analysis, our relatively small sample (43 participants, 86 legs) may limit the generalizability of our findings. Despite implementing methodological safeguards against overfitting (including five-fold cross-validation, maintaining a separate test set, and using multiple performance metrics), caution is warranted when interpreting the results. In addition, due to the relatively small dataset size, we were unable to create a separate validation set beyond our training and test sets. This three-way split (training, validation, and test sets) is considered ideal practice in machine learning as it allows for unbiased hyperparameter optimization on the validation set before final evaluation on the test set. Larger, more diverse cohorts should be examined to validate and refine these models. Third, our 2D video analysis methodology cannot capture out-of-plane movements, which may result in measurement errors when movements occur outside the primary recording plane. Additionally, perspective errors and marker occlusion can affect measurement accuracy. Despite these limitations, we selected 2D video analysis for its greater clinical applicability and feasibility compared to 3D motion capture, which typically requires specialized laboratory settings, expensive equipment, and technical expertise. Fourth, focusing only on frontal plane kinematics may not capture the full complexity of EOA-related movement patterns. Future studies should incorporate multiplane analysis and additional biomechanical variables (e.g., kinetics and muscle activation) for a more comprehensive understanding of EOA-related movement patterns. Fifth, while the EOAQ was developed by an international expert panel specifically for early OA detection, comprehensive validation studies establishing its sensitivity and specificity have not yet been published. The EOAQ represents one of the first standardized instruments designed specifically for detecting early-stage knee OA before significant structural changes are evident on conventional imaging, making it suitable for our research question. However, future studies would benefit from correlating EOAQ results with other clinical and imaging biomarkers to establish its diagnostic accuracy.

Future research should expand upon our current methodology in several key directions. First, deep learning models (such as convolutional neural networks and recurrent neural networks) could be implemented to potentially improve classification performance and automatically extract relevant features from raw kinematic data without the need for manual feature engineering. Second, investigating the ability of machine learning models to distinguish between different severity levels of EOA would provide valuable clinical insights for disease progression monitoring and treatment planning. Third, incorporating different view variations (including sagittal and transverse planes) would offer a more comprehensive three-dimensional analysis of movement patterns associated with EOA. Finally, developing techniques to address self-occlusion during movement analysis would improve the reliability of kinematic measurements, particularly in complex functional tasks. These advancements would collectively enhance the clinical applicability and diagnostic accuracy of machine learning-based EOA detection using kinematic data.

Conclusion

This study demonstrated the potential of ML analysis of frontal plane kinematics during SU and SD tests to detect and classify EOA among recreational table tennis players. These findings contribute to a growing body of evidence supporting the use of functional biomechanical assessments in EOA diagnosis and management. While these results are promising, they should be considered preliminary given our relatively small and specific study population. Future research should focus on validating these findings in larger, more diverse cohorts, investigating the longitudinal changes in kinematic patterns during EOA progression, and exploring the integration of kinematic data with other clinical and imaging biomarkers to enhance the detection and prognosis of EOA.

Supplemental Information

Supplemental Information 1 Raw data

Additional Information and Declarations

Competing Interests

Author Contributions

Human Ethics

Data Availability

The authors declare there are no competing interests.

Ui-jae Hwang conceived and designed the experiments, performed the experiments, analyzed the data, prepared figures and/or tables, authored or reviewed drafts of the article, and approved the final draft.

Kyu Sung Chung performed the experiments, analyzed the data, prepared figures and/or tables, and approved the final draft.

Sung-min Ha conceived and designed the experiments, performed the experiments, prepared figures and/or tables, authored or reviewed drafts of the article, funding, and approved the final draft.

The following information was supplied relating to ethical approvals (i.e., approving body and any reference numbers):

The present study conformed to the ethical guidelines of the 1975 Declarations of Helsinki. This study was approved by the Sangji University Institutional Review Board (1040782-230814-HR-09-117). Informed consent for publication of the images was obtained from the patient.

The following information was supplied regarding data availability:

The raw measurements are available in the Supplementary File.

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
