# Peer review of "Clustering and classification of early knee osteoarthritis using machine-learning analysis of step-up and down test kinematics in recreational table tennis players"

_PeerJ, doi:10.7717/peerj.19471_

## Round 0.1 · original submission · Major Revisions

Some revisions are needed before acceptance.

**Language Note:** The review process has identified that the English language must be improved. PeerJ can provide language editing services - please contact us at [email protected] for pricing (be sure to provide your manuscript number and title). Alternatively, you should make your own arrangements to improve the language quality and provide details in your response letter. – PeerJ Staff

·

Basic reporting

1. The English of the article can be improved, and grammar should be reanalyzed.
2. Long sentences should be avoided to make them more convenient and understandable for the reader. For example: lines 316-318.
3. The use of the word "we" seems to be used again and again in the article. Those sentences should be reframed as the use of the word "we" does not sound professional. For example: In lines: 234, 239, and many more.
4. The article sections need to be restructured to avoid confusion. For Example, the experimental protocol should be placed before section 2.3. Don't make the ML section 2.6. Despite, make it section 3 and then 3.1.... etc. onwards.
5. The caption of Figure 1 should be improved to depict its clear meaning (it indicates the methodology to implement the strategy, not just flow).
6. A proper hypothesis should be formed that needs to be checked and based on the results, it should be accepted or rejected.
7. In the Introduction section, add a line indicating the relevance of OA based on WHO's recent data.
8. In the Introduction section, the authors talked about the limitations of conventional methods in OA detection but did not highlight any (E.g. high cost and time, faulty results, expert requirement, etc.) to justify the need for ML-based models.
9. Despite using machine learning multiple times, make its abbreviation as ML.
10. Authors can add an acquisition setup diagram to depict the data collection process. For reference, you can follow the paper in the given link (https://link.springer.com/article/10.1007/s11042-024-19673-z)
11. Make the results section as: statistical analysis & ML-based analysis.
12. In the abstract section:
(i) Some lines indicating the drawbacks in the related past research should be added, based on which the objective of this study is designed.
(ii) The Methods part needs improved writing.
(iii) In the results part, add p-value and ML model's performance in %age to make it more robust (i.e. add numerical data).

Experimental design

1. How the landmarking of anatomical points has been performed? (Automatic or manual). Please Elaborate and add sufficient details.
2. Deep Learning seems to have improved results than ML in recent years. Why did the authors opt to perform an analysis of OA using ML, but not DL? Any specific reason?
3. The authors used metrics like F1 score, precision, etc. but did not elaborate on them. These metrics values should also be added to the results in the writeup.
4. Add more details on the statistical technique utilized to check the significance of variables (p-value).
5. The data preparation and processing section is not clear. what data is processed and what techniques have been utilized? Add relevant details

Validity of the findings

The authors analyzed the data considering different variables which is adequate. The results are well supported by the literature studies. Further, the following points can be added:

1. A graph showing the comparative analysis of results achieved using all the employed ML methods to depict the best one.
2. The validation of the hypothesis should be indicated (accept/reject) based on the results of p-values.
3. The validation of ML model results (such as considering accuracy) can be performed using some statistical test such as Friedman's rank test.
4. A table may be formed showing the comparison of the current study with the state-of-the-art based on performance metrics.

Additional comments

Others points such as:
Experimentation using DL models, Severity levels, Different view variations, Self-occlusion
can be added to be performed in future work.

Reviewer 2 ·

Basic reporting

Thank you for submitting your manuscript on the clustering and classification of early knee osteoarthritis (EOA) using machine-learning analysis of step-up and step-down test kinematics in recreational table tennis players. Your study presents a novel approach to EOA detection using machine-learning techniques applied to functional movement assessments. However, there are several methodological, statistical, and interpretative issues that need to be addressed to enhance the clarity, rigor, and impact of the study.

#1. The objective is not clearly defined. The terms "classification" and "prediction" are used interchangeably without clear distinction.
Recommendation: Clarify whether the study aims to classify different EOA severity levels, predict its progression, or simply differentiate between EOA and non-EOA cases. Provide specific numerical results (e.g., AUC values) to substantiate the claims of "high performance."
#2. The novelty of the study is not explicitly highlighted. There is insufficient discussion on how this study differs from prior work using motion capture, 3D gait analysis, or wearable sensors.
Recommendation: Clearly articulate the unique contribution of using 2D video analysis compared to existing biomechanical assessment methods. Provide comparative references to previous research.
#3. The justification for selecting recreational table tennis players as the study population is weak. The claim that table tennis increases the risk of OA lacks supporting epidemiological data.
Recommendation: Provide references or epidemiological data demonstrating that table tennis players are at increased risk of OA compared to other sports or the general population.
#4. The study includes only 43 participants (86 legs), which raises concerns regarding the robustness of the machine-learning models.
Recommendation: Provide a justification for the sample size with a power analysis or discuss potential overfitting and limitations due to a small dataset.
#5. The validity and reliability of the Early Osteoarthritis Questionnaire (EOAQ) are not adequately discussed.
Recommendation: Include sensitivity, specificity, and previous validation studies of EOAQ, if available.
#6. The limitations of 2D video analysis, such as its inability to capture out-of-plane movements, are not discussed.
Recommendation: Acknowledge these limitations and justify why 2D video was chosen over 3D motion capture systems.
#7. The description of model hyperparameters, cross-validation techniques, and overfitting prevention strategies is insufficient.
Recommendation: Provide details on hyperparameter tuning, feature selection methods, and cross-validation techniques used to ensure model robustness.
#8. The Louvain clustering results lack validation metrics such as silhouette scores or explained variance ratios.
Recommendation: Provide cluster validity indices to support the claim that four clusters are meaningful and distinct.
#9. The reported AUC values (e.g., 1.000 for test data) suggest possible overfitting.
Recommendation: Perform an external validation with an independent dataset or discuss potential overfitting concerns.
#10. Feature importance analysis is described but lacks biomechanical interpretation.
Recommendation: Explain why features such as AHD-SU and FHD-SD are the most predictive and their relevance in OA pathophysiology.
#11. The study does not adequately compare its performance with conventional OA diagnostic methods such as MRI, radiography, or gait analysis.
Recommendation: Discuss how this method compares to traditional diagnostic tools and its potential clinical applications.
#12. The discussion does not adequately explain why certain kinematic parameters were significant predictors of EOA.
Recommendation: Provide a biomechanical rationale for why horizontal ankle and femur displacement are the strongest predictors.
#13. The discussion lacks a thorough examination of study limitations.
Recommendation: Expand the discussion on potential biases, the need for external validation, and future research directions.
#14. The conclusion overgeneralizes the applicability of the findings, given the small and specific study population.
Recommendation: Clarify that while the findings are promising, further validation in larger, diverse cohorts is required.

Experimental design

no comment

Validity of the findings

no comment

Additional comments

no comment

---

## Round 0.2 · accepted · Accept

The authors have sufficiently addressed the reviewers' comments and suggestions. The paper can be accepted now.